# Characterization of the neurogenic niche in the aging dentate gyrus using iterative immunofluorescence imaging

John Darby Cole[1], Jacobo Sarabia del Castillo[2], Gabriele Gut[2], Daniel Gonzalez-Bohorquez[1], Lucas Pelkmans[2], Sebastian Jessberger[1]*

[1]Brain Research Institute, University of Zurich, Zurich, Switzerland; [2]Department of Molecular Life Sciences, University of Zurich, Zurich, Switzerland

**Abstract** Advancing age causes reduced hippocampal neurogenesis, associated with age-related cognitive decline. The spatial relationship of age-induced alterations in neural stem cells (NSCs) and surrounding cells within the hippocampal niche remains poorly understood due to limitations of antibody-based cellular phenotyping. We established iterative indirect immunofluorescence imaging (4i) in tissue sections, allowing for simultaneous detection of 18 proteins to characterize NSCs and surrounding cells in 2-, 6-, and 12-month-old mice. We show that reorganization of the dentate gyrus (DG) niche already occurs in middle-aged mice, paralleling the decline in neurogenesis. 4i-based tissue analysis of the DG identifies changes in cell-type contributions to the blood-brain barrier and microenvironments surrounding NSCs to play a pivotal role to preserve neurogenic permissiveness. The data provided represent a resource to characterize the principles causing alterations of stem cell-associated plasticity within the aging DG and provide a blueprint to analyze somatic stem cell niches across lifespan in complex tissues.

*For correspondence:
jessberger@hifo.uzh.ch

## Introduction

Somatic stem cells divide throughout life to generate progeny that is required for tissue homeostasis and repair (*Fuchs and Segre, 2000*). With advancing age, there is a decline in stem cell numbers, output, and function, which has been associated to altered organ integrity and reduced tissue repair in aging mammals (*Ermolaeva et al., 2018*). In addition to age-dependent, cell-intrinsic alterations causing reduced functionality of somatic stem cells, there is ample evidence that changes in the surrounding stem cell niche contribute to age-related stem cell phenotypes (*Morrison and Spradling, 2008*; *Dulken et al., 2019*; *Navarro Negredo et al., 2020*). However, the exact spatial relationships between altered niche cells and somatic stem cells remain incompletely understood, likely because conventional immunohistochemical approaches only allow for the simultaneous detection of few proteins. Further, conventional RNA sequencing approaches or proteome analyses do not allow for the recovery of spatial information within complex tissues. Thus, recent work aimed to overcome these limitations, for example by the development of spatial transcriptomics, multiplexed protein detection, and mass-spectrometry based approaches allowing for the detection of dozens of proteins with cellular resolution (*Giesen et al., 2014*; *Ståhl et al., 2016*; *Schapiro et al., 2017*; *Lin et al., 2018*; *Schulz et al., 2018*; *Burgess, 2019*; *Moncada et al., 2020*). Based on these technological innovations major advances have been achieved characterizing the cellular organization and complexity of mouse and human tissues in health and disease (*Decalf et al., 2019*; *Wagner et al., 2019*). However, these approaches largely miss three-dimensional (3D) information in tissues and require highly specialized equipment and/or customized or conjugated antibodies, reducing their availability. We recently developed the 4i (iterative indirect immunofluorescence imaging) technology, which allows for the detection

of >40 individual proteins or protein states using conventional primary and secondary antibodies and mild conditions of antibody elution by preventing photo-induced crosslinking during imaging, thereby preserving sample integrity and achieving high reproducibility at very high spatial resolution across many cycles (*Gut et al., 2018*). Since 4i was originally established for monolayers of cultured cells, we here developed a robust protocol to apply 4i to histological tissue sections to characterize the aging dentate gyrus (DG) niche of the mouse hippocampus, one of the brain areas where neural stem cells (NSCs) generate new neurons throughout life (*Gage, 2019*). Advancing age has been associated with strongly reduced NSC activity and diminished neuronal output, which is due to cell-intrinsic, niche-dependent, and humoral mechanisms (*Villeda et al., 2011*; *Moore et al., 2015*; *Boldrini et al., 2018*; *Kalamakis et al., 2019*; *Moreno-Jiménez et al., 2019*; *Tobin et al., 2019*; *Denoth-Lippuner and Jessberger, 2021*). Using 4i, we show that the cellular reorganization of the DG niche occurs relatively early in life and identify substantial alterations in contact sites between NSCs and the vasculature, and niche-associated changes that are paralleled with a decline in neurogenesis. The data presented here characterize age-related changes in the DG niche and provide a blueprint for the analyses of a substantial number of proteins in single cells and their surrounding niche within complex tissues.

## Results

### Iterative 4i immunostaining in complex mouse and human tissues

To achieve multiple rounds of 4i on tissue, sections needed to be adequately mounted to minimize loss of samples through detachment without affecting antigenicity. Coating the glass-bottomed well-plates with poly-d-lysine (PDL) was effective to preserve tissue adhesion across multiple cycles of 4i in sections of 3-month-old adult mouse brain (*Figure 1A*), human embryonic stem cell (hESC)-derived regionalized forebrain organoids (*Figure 1B*), and embryonic day 14.5 (E14.5) mouse brain (*Figure 1C*). Thus, using this approach, sequential rounds of conventional antibody stainings in mouse and human tissues becomes achievable (*Figure 1A–C*, for details of used antibodies refer to the legend of *Figure 1*). Notably, 4i in tissue sections allows for 3D reconstruction of tissue volumes, facilitating detailed spatial analyses throughout tissue sections (*Figure 1D*), showcased using antibodies against IBA1-labeled microglia, Nestin- and GFAP-labeled NSC processes, DCX-expressing immature neurons, LaminB1-expressing cells, and CollagenIV-labeled vasculature (*Figure 1D*). To assess potential effects of cyclic staining and repeated elutions on sample antigenicity, adult brain sections were stained with the 18 antibodies used in the present study, imaged, and subsequently subjected to six rounds of elution prior to restaining. Measured fluorescence intensities between the stainings before and after rounds of elution were strongly correlated (*Figure 1E* and *Figure 1—figure supplement 1*), indicating that antigenicity is preserved across repeated cycles of iterative immunostainings for most of the antibodies that we used in tissue sections, similar to the high reproducibility that 4i achieves in cultured cells (*Gut et al., 2018*). Significant correlation was not achieved for the stem cell marker ID4 and MT3+ cells were poorly correlated between rounds. While staining was impaired for ID4, the quality was improved for MT3, suggesting the elution step may have some efficacy for antigen retrieval.

### Cellular alterations in the DG niche with advancing age

After establishing a robust protocol for iterative immunostainings in complex tissues, we used 4i to analyze age-associated cellular and molecular alterations in the DG of the hippocampal formation. NSCs generate new neurons throughout life in the DG. However, hippocampal neurogenesis is significantly decreasing during adulthood (*Ben Abdallah et al., 2010*; *Denoth-Lippuner and Jessberger, 2021*). To test for 4i's ability to identify age-related changes within the DG, a panel of commonly used antibodies was applied iteratively on the exact same sections to visualize a number of diverse cell types in 2-, 6-, and 12-month-old brain sections: 'Stem cells' were visualized using HOPX, Nestin, GFAP, SOX2, ID4, MT3, and LaminB1; 'Proliferation' was assessed using pH3 and KI67, 'Neurons' were labeled with DCX, PV, ARC (i.e., expressed after plasticity-inducing activity), and LaminB1, 'Glial cells' were stained for NG2, OLIG2, IBA1, S100β, and GFAP, 'Vasculature' was analyzed with CD13 and Collagen IV (*Figure 2A–D* and *Figure 2—figure supplements 1–4*; *Ben Abdallah et al., 2010*; *Denoth-Lippuner and Jessberger, 2021*). Radial glia-like stem cells (R, also referred to as type 1 cells) were phenotyped based on the combinatorial expression of HOPX, SOX2, and the extension of a glial

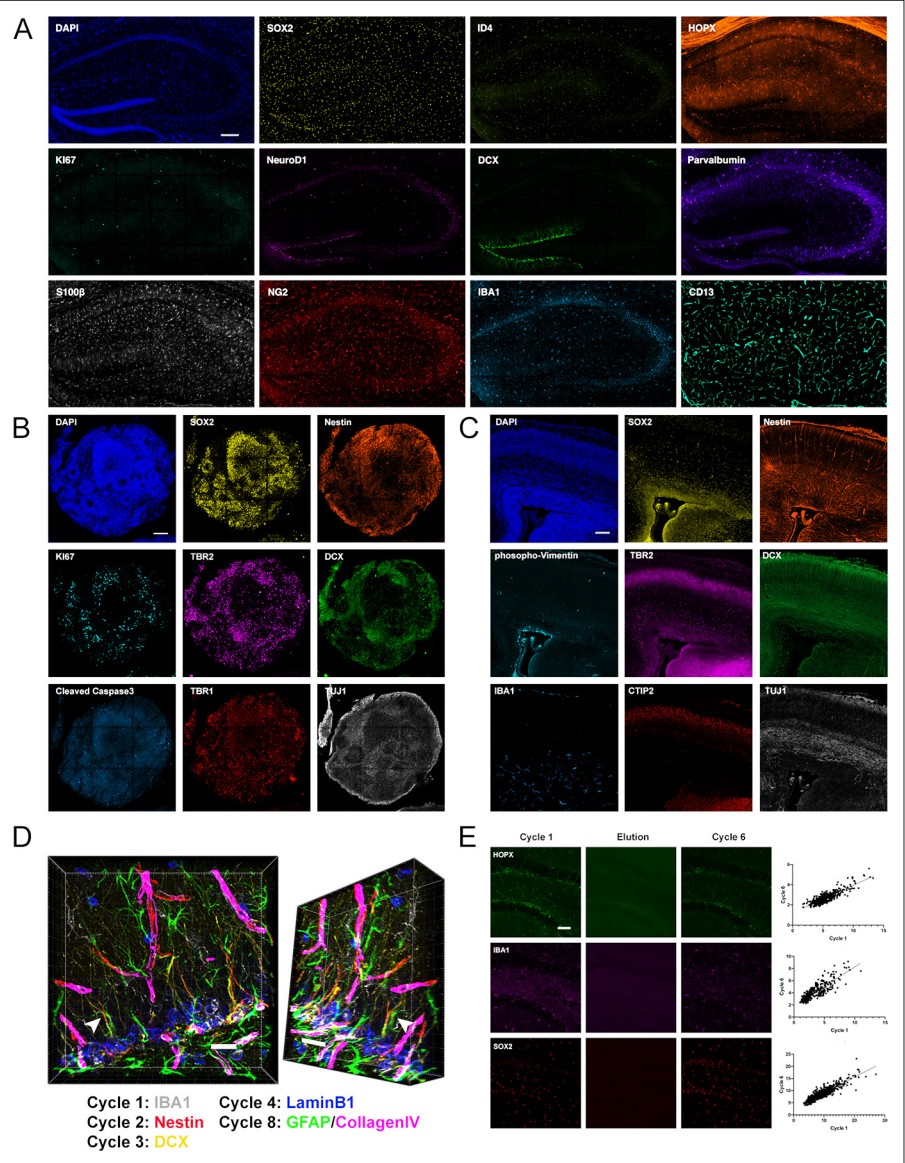

**Figure 1.** Iterative immunostaining in mouse and human tissue sections. (**A**) Shown is an adult mouse dorsal hippocampal section labeled with 11 antibodies, as indicated in the individual panels, acquired over five rounds of 4i. SOX2/ID4/HOPX astrocytes/NSCs; KI67, proliferation; NeuroD1/DCX, immature neurons; Parvalbumin, interneurons; S100β, astrocytes; NG2, oligodendroglia; IBA1, microglia; and CD13, pericytes. Nuclei were counterstained with DAPI. (**B**) An hESC-derived, forebrain organoid section fixed at day 40 in vitro and labeled with eight antibodies, as indicated in the individual panels, over four rounds of 4i. SOX2/Nestin, apical progenitors; KI67, proliferation; TBR2, basal progenitors; DCX, immature neurons; cCaspase3, apoptotic cells; and TBR1/TUJ1, neurons. Nuclei were counterstained with DAPI. (**C**) Shown is a cortical section from an E14.5 mouse embryo labeled with eight antibodies, as indicated in the individual panels, over four rounds of 4i. Phopho-Vimentin, intermediate filaments; IBA1, microglia; SOX2/Nestin, astrocytes/NSCs; TBR2, basal progenitors; CTIP2, neurons; and DCX/TUJ1, immature neurons. Nuclei were counterstained with DAPI. (**D**) 3D reconstruction of a region in the dorsal DG analyzed with antibodies as indicated, with an arrow highlighting a Nestin+/GFAP+ R cell radial process indicating the spatial fidelity of 4i across cycles. (**E**) Examples of HOPX, IBA1, and SOX2 labeling in adult mouse brain sections, with proof of elution efficacy and restaining quality. The normalized fluorescent intensities for each stain were correlated between rounds. For details of statistics please refer to ***Supplementary file 1***. Scale bars represent 200 μm (**A**), 100 μm (**B, C**), 25 μm (**D**), and 50 μm (**E**). 4i, immunofluorescence imaging; DG, dentate gyrus; hESC, human embryonic stem cell.

The online version of this article includes the following figure supplement(s) for figure 1:

*Figure 1 continued on next page*

*Figure 1 continued*

**Figure supplement 1.** Examples of used antibody labeling in adult mouse brain sections, with proof of elution efficacy and restaining quality.

fibrillary acidic protein (GFAP) positive radial process (***Denoth-Lippuner and Jessberger, 2021***). As HOPX, SOX2, and GFAP are also expressed in astrocytes, only cells that were negative for the astrocytic calcium-binding protein S100β, a marker for mature astrocytes, were considered to be R cells. Proliferating non-radial (NR) cells were identified in the subgranular zone (SGZ) by the expression of KI67 and absence of a GFAP-labeled radial process. Immature neurons were classified based on doublecortin (DCX) expression. Antibodies were used in the same order for all age groups analyzed (***Figure 2—figure supplement 1B***).

Using the whole panel of antibodies, distinct cell types were quantified in the SGZ and normalized to the volume of the DG granule cell layer (GCL) (***Figure 3A–C***). As expected, the density of R cells was significantly reduced in animals with advancing age (***Figure 3A–B***; ***Denoth-Lippuner and Jessberger, 2021***). While the total density of proliferating KI67$^+$ cells decreased with advancing age, KI67$^+$ R cells did not change. Additionally, the density of immature neurons, positive for DCX was strongly reduced at 12 months of age (***Figure 3A–B***).

In the neurogenic niche, R cells interact with and are regulated by several different cell types that are present within their DG niche (***Song et al., 2016***; ***Mosher and Schaffer, 2018***). To assess potential changes in the cytoarchitecture of the neurogenic niche, microglia, astrocytes, and oligodendrocytes were counted and normalized to the total tissue volume as well as the sub-regions of the DG, the molecular layer (ML), GCL, and hilus. There were no significant differences in IBA1-labeled microglia, or oligodendrocyte precursor cells (OPCs, expressing OLIG2 and NG2) in the DG or within the analyzed DG sub-regions (***Figures 2A–B , and 3C***). However, in animals of advanced age, the numbers of astrocytes, expressing S100β, in the whole DG were reduced (***Figure 3A and C***).

We next used 4i to analyze the expression levels of selected proteins in the same R cells identified in the quantification of cell density. We focused on HOPX and MT3, which have been identified to be present in R cells and may be related to R cell quiescence (***Shin et al., 2015***; ***Berg et al., 2019***; ***Bottes et al., 2021***). Supporting previous data suggesting that advancing age is associated with increased R cell quiescence (***Kalamakis et al., 2019***; ***Harris et al., 2021***), we identified in 12-month-old animals increased fluorescent intensities of HOPX and MT3 (***Figure 4A and B***). Notably, altered expression of HOPX and MT3 was not observed in classical astrocytes classified via expression of S100β (***Figure 4C***).

To further validate the 4i approach on tissue sections, we analyzed age-related expression of LaminB1 (LB1), one of the four intermediate filaments that make up the nuclear lamina, and where reductions with aging and the premature aging phenotype of progeria have been previously described (***Burke and Stewart, 2013***). Confirming previous reports, we found reduced levels of LB1 between 2- and 12-month-old animals in R cells and DCX-labeled immature neurons (***Bedrosian et al., 2021***; ***Bin Imtiaz et al., 2021***). As 4i allows for analyzing expression levels in other cell types within exactly the same sections, we found that microglia, astrocytes, and Parvalbumin (PV)-labeled interneurons in the niche showed no significant changes with age in their LB1 expression levels (***Figure 4D and E***).

## Decline of vascular network and alterations of BBB composition with age

In the DG, both R and NR cells have close relationships with surrounding blood vessels that are the source of regulatory trophic factors, originating from vascular endothelial cells, and connect the neurogenic niche to the circulating blood system. Across species, the hippocampus shows a strong age-related decline in vascularization (***Katsimpardi et al., 2014***; ***Boldrini et al., 2018***). 4i-based analysis confirmed that vascular density decreases with age, with reductions already visible by 6 months of age (***Figure 5A–B***). In addition to declining vascular networks, age has been associated with an increase in leakiness of the blood-brain barrier (BBB), potentially permitting molecules into the brain leading to increased inflammation and neurotoxicity (***Bell et al., 2010***). BBB integrity was estimated by measuring the percent of vasculature covered by other cell types based on signal colocalization. While there was no significant change in the total percentage of vasculature coverage, differences were observed in the contributions of different cell types (***Figure 5C–D***). Pericytes, expressing CD13 and representing one of the primary cell-types that make up the BBB (***Bell et al., 2010***), were

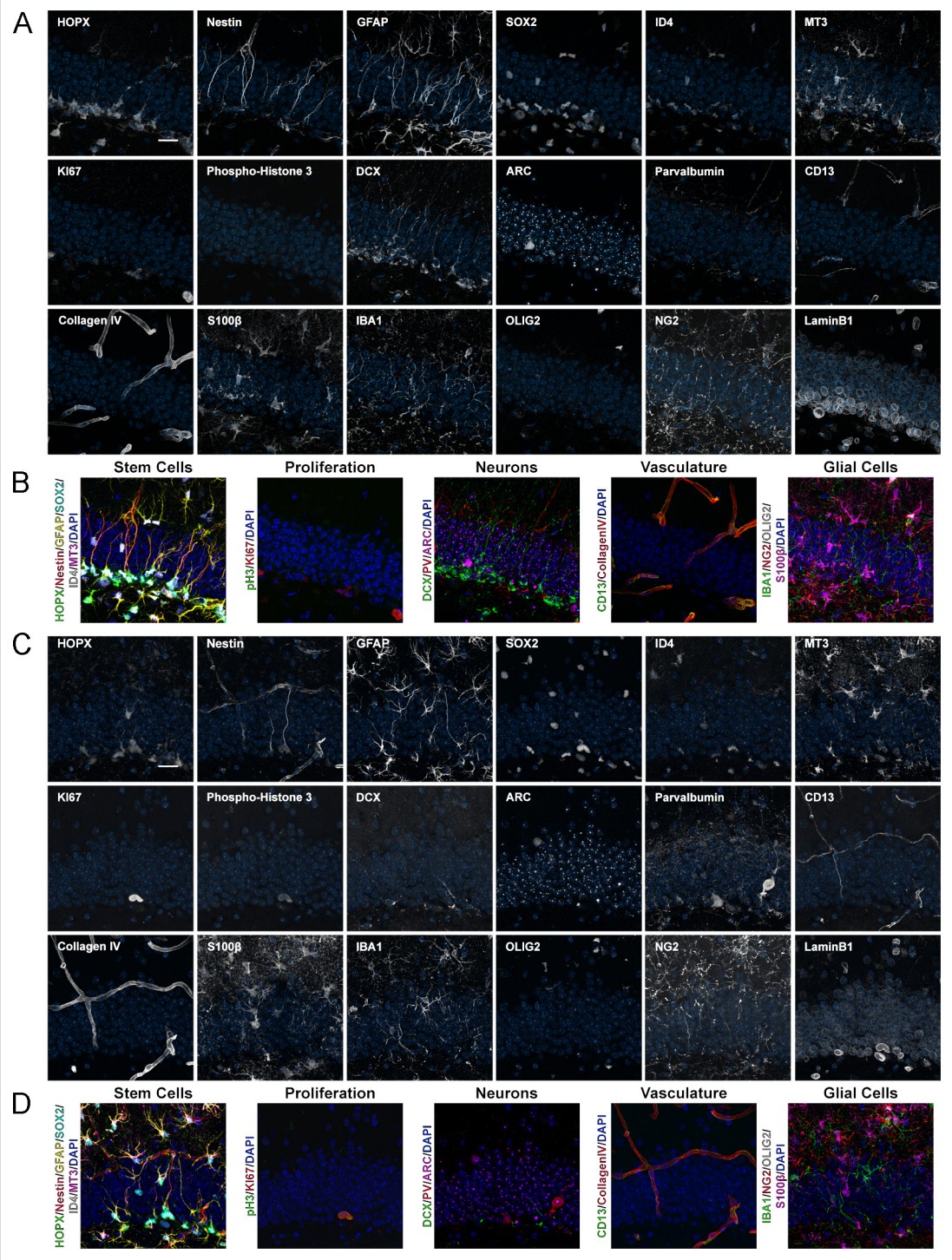

**Figure 2.** Expression of 18 proteins in the DG of mice with different ages. (**A**) Images of 18 proteins labeled in the same area of the DG from a 2-month-old mouse. HOPX/Nestin/GFAP/SOX2/ID4/MT3 astrocytes/NSCs; KI67/phospho-Histone 3, proliferation; DCX, immature neurons; ARC, mature neurons; Parvalbumin, interneurons; CD13, pericytes; CollagenIV, vasculature; S100β, astrocytes; IBA1, microglia; OLIG2/NG2, oligodendroglia; and LaminB1, nuclear lamina. (**B**) shows merged channels for stem cells, proliferation, neurons, vasculature, and glial cells. Nuclei were counterstained with DAPI. (**C**) Images of 18 proteins labeled in the same area of the DG from a 12-month-old mouse. (**D**) shows composites grouping markers based on cell type. Note that the same sections were used for *Figure 4*, highlighting the power of iterative immunostaining using 4i. Scale bars represent 25 μm. DG, dentate gyrus; NSC, neural stem cell.

The online version of this article includes the following figure supplement(s) for figure 2:

*Figure 2 continued on next page*

*Figure 2 continued*

**Figure supplement 1.** Expression of 18 proteins in the DG of 6-month-old mice.

**Figure supplement 2.** Overview expression of 18 proteins in the DG of 2-month-old mice.

**Figure supplement 3.** Overview expression of 18 proteins in the DG of 6-month-old mice.

**Figure supplement 4.** Overview expression of 18 proteins in the DG of 12-month-old mice.

significantly reduced at 6 months with area covered remaining stable at 12 months. Interestingly, there appeared to be a compensatory increase in coverage by astrocytes labeled by GFAP (*Figure 5C–D*). GFAP-expressing cells may also include Nestin$^+$ R cells; however, their contribution to increased GFAP vasculature coverage with advancing age appears unlikely given the substantially reduced coverage of Nestin-expressing cells (*Figure 5D*). Additionally, there was a strong trend for an increase in microglia-blood vessel contact at 6 months that reached significance in 12-month-old mice compared to young adult mice (*Figure 5C–D*). The contribution of R cells (expressing Nestin) to vasculature coverage significantly decreased with age, likely due to the decrease in population, while OPC (labeled with NG2) contact sites with the vasculature remained consistent at each age point (*Figure 5C–D*).

## Microniche-based analysis identifies novel age-related characteristics of the DG

We speculated that within the aging DG neurogenic niche, microenvironments may exist possessing distinct capacities for preservation of neurogenic processes. Spots were randomly distributed across the GCL spaced 50 μm apart. Utilizing the multidimensionality of the data set acquired with 4i, volumes of 11 cell markers were measured within a 50 μm radius of each spot to achieve contiguous sampling of 'microniches' in the GCL and bordering areas of the hilus and ML (*Figure 6A*). The aim of such microniche-based analyses was to identify age-related changes within the immediate proximity of neurogenic cells. Post hoc analysis showed significant differences (in terms of percent of marker expression per randomized microniche volume) in a variety of measured cell types within microniches (*Figure 6B*). For example, vascular and pericyte densities were similarly decreased at 6 and 12 months compared to 2-month-old mice. While the stem cell marker Nestin was significantly reduced by 6 months, GFAP, which is also present in classical astrocytes, remained stable between each age time point, indicating that classical astrocytes contribute substantially to GFAP-expression volumes in analyzed microniches (*Figure 6B*). Additionally, SOX2 was reduced at 6 months, and decreased further in 12-month-old animals. For glial cells, IBA1-expressing microglia sequentially increased with age while OPCs showed a temporary increase at 6 months with a trending decrease at 12 months (*Figure 6B*). Thus, 4i applied to tissue sections allows for the identification of cell-type specific alterations with advancing age.

For dimensional reduction, principal component analysis was used to cluster the sample regions based on their cellular contents. The majority of regions from 2-month-old animals formed a cluster that was distinct from 6- and 12-month-old animals, which primarily clustered together (*Figure 6C*). As the spots were largely segregated into two groups, k-means clustering was performed expecting two populations; one 'young-like' and one 'old-like' identified based on neurogenic cell content. In the 2-month-old animals, 64.5% of spots were grouped as young while 35.5% were labeled as old (*Figure 6D*). Comparing the contents of the two classes of spots, 'young' spots showed higher volumes of Nestin, SOX2, KI67, DCX, CollagenIV, PV, IBA1, and NG2 (*Figure 6E*). As such, the development of an 'aged' phenotype occurred for select niches already at 2 months of age and became already very pronounced in 6-month-old mice. To explore the potential of microenvironments that may support the persistence of R cells within the DG, spots were split by Nestin radial process content. Spots containing a population of Nestin-labeled R cells were classified as high Nestin-containing areas when they were falling within the top three quartiles of spots identified in young mice ( 0.4% volumetric coverage) (*Figure 6—figure supplement 1a*). At 2 months, there were no differences in the volumes of CollagenIV vasculature or IBA1-expressing microglia between high and low Nestin populated spots (*Figure 6F*). However, in both the 6- and 12-month-old groups, both CollagenIV and IBA1 were robustly elevated in the 'high' Nestin spots (*Figure 6F*). In contrast, spots containing Ki67$^+$ proliferating cells, CollagenIV and IBA1 volumes were comparable to Ki67$^-$ spots from middle-aged and aged animals (*Figure 6B–D*). Thus, 4i identified preferential occupancy of blood vessels and

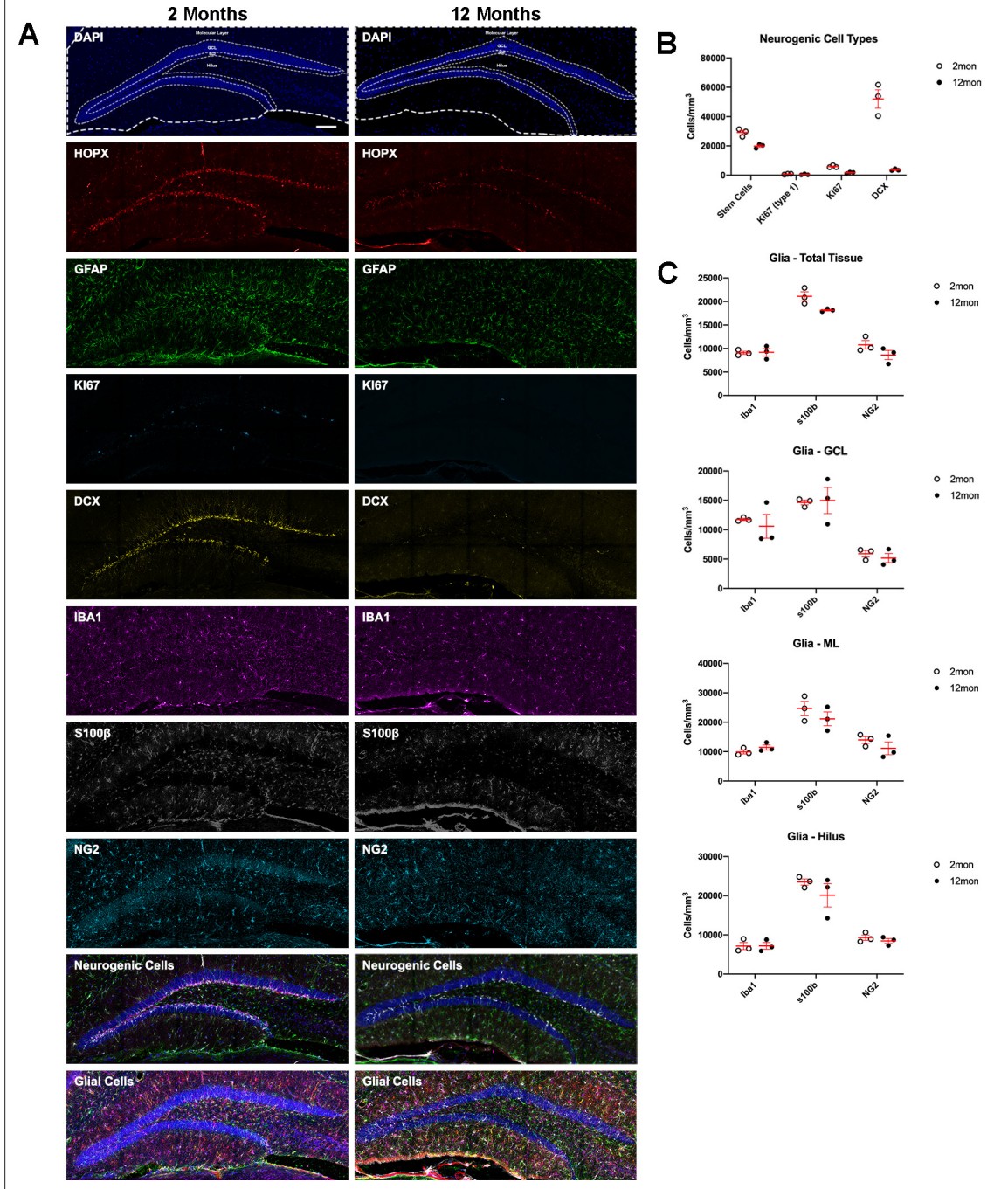

**Figure 3.** Age-related dynamics of cell populations in the mouse DG. (**A**) Example overviews of the same DGs from 2- and 12-month-old mice showing the neurogenic and glial markers used to phenotype cells in population density measures. Nuclei were counterstained with DAPI. (**B**) Density analyses of neurogenic cells in the SGZ, defined as the number of R cells. For details of statistics please refer to *Supplementary file 1*. (**C**) Density analyses of microglia, OPCs, and astrocytes in the total sampled area of the DG, in the granule cell layer, the molecular layer, and hilus. For details of statistics please refer to *Supplementary file 1*. Scale bar represents 100 μm. *p<0.05, **p<0.01. DG, dentate gyrus; OPC, oligodendrocyte precursor cell; SGZ, subgranular zone.

The online version of this article includes the following figure supplement(s) for figure 3:

**Figure supplement 1.** Expression of stem cell and glial cell markers in the DG.

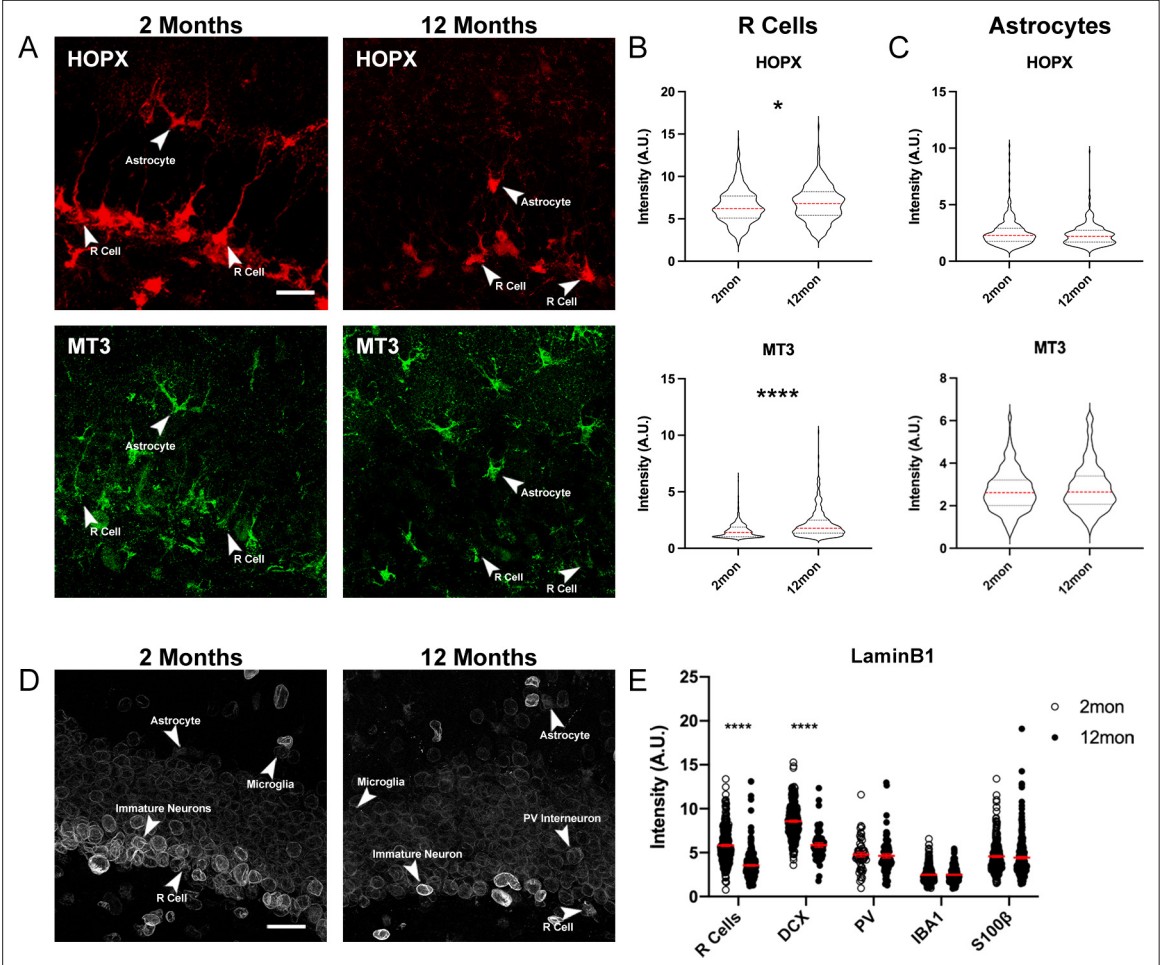

**Figure 4.** Age-dependent changes of protein expression in the DG. (**A**) Representative images of 2- and 12-month-old DG sections labeled with quiescent markers HOPX and MT3. Arrowheads indicate the same cell across stains. (**B**) Quantification of the fluorescent intensities of HOPX and MT3 in R cells normalized to background. (**C**) Quantification of the fluorescent intensities of HOPX and MT3 in astrocytes normalized to background. For details of statistics please refer to *Supplementary file 1*. (**D**) Images showing LaminB1 in DGs from 2- and 1 12-month-old mice with different cell types indicated by arrows. (**E**) Quantification of the fluorescent intensities normalized to background. For details of statistics please refer to *Supplementary file 1*. Scale bars represent 25 μm. ****p<0.0001. DG, dentate gyrus.

IBA1-labeled cells in Nestin-containing microniches, indicating a potential role for the vasculature and microglial cells in the maintenance of quiescent stem cells.

## Age-dependent dynamics of potential stem cell contact sites in the DG niche

Taking advantage of the ability to analyze 3D cellular interactions using 4i on complex tissues, we next characterized potential contact sites (as judged by proximity) of radial processes extending from R cells (*Seri et al., 2001*; *Kronenberg et al., 2003*). While little is still known about the function of the radial processes, they are believed to have a role in receiving regulatory signals from the surrounding niche (*Moss et al., 2016*). To analyze possible interactions between R cells and other cell types in the niche, areas of niche cell marker colocalization with Nestin+ RGL processes were measured and normalized to the area of RGL processes. While the optical resolution was not sufficient to definitively identify contact points, signal colocalization indicates a proximity of cells within ~1 μm. In general, a greater percentage of radial processes in 12-month-old animals were in close proximity with niche cell markers for vasculature, pericytes, and microglia, than in younger mice (*Figure 7A–B*). R cell processes were preferentially close to the vasculature in areas with low pericyte coverage (areas that are negative for CD13) (*Licht et al., 2020*), but colocalization of radial processes with CD13-labeled

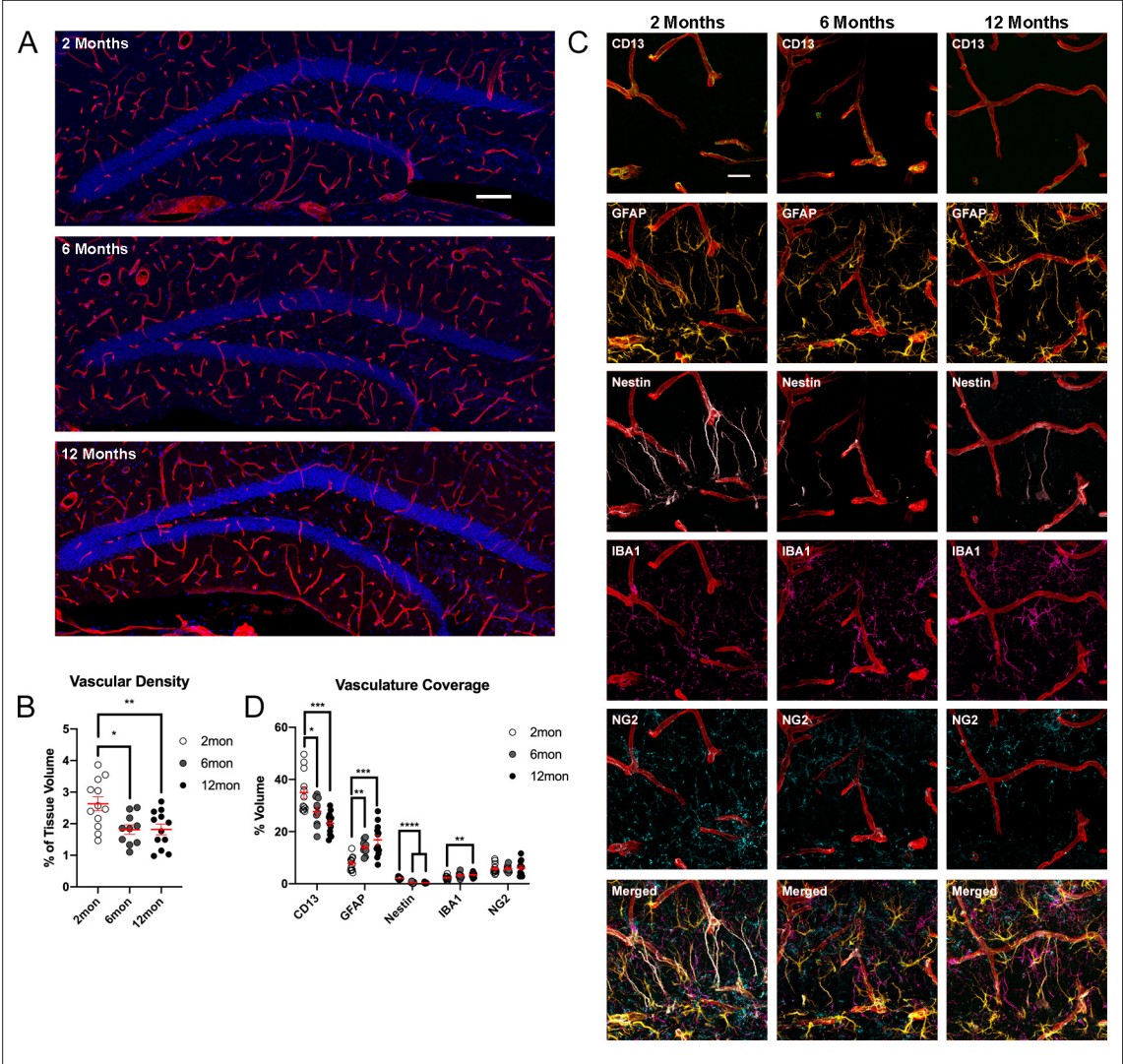

**Figure 5.** Changes of DG vasculature with advancing age. (**A**) Images of CollagenIV+ blood vessels in the DGs at 2, 6, and 12 months. Nuclei were counterstained with DAPI. (**B**) Quantification of vascular density in the DGs of 2-, 6-, and 12-month-old mice. For details of statistics please refer to *Supplementary file 1*. (**C**) Representative images showing the interactions of pericytes, glia, and R cells with the vasculature in the DG. (**D**) Quantification of colocalization of pericytes, astrocytes, R cells, microglia, and OPCs with CollagenIV+ represented a percentage of total vascular volume. For details of statistics please refer to *Supplementary file 1*. Scale bars represent 100 μm (**A**), 25 μm (**C**). **p<0.01, ***p<0.001, ****p<0.0001. DG, dentate gyrus; OPC, oligodendrocyte precursor cell.

pericytes was increased at 12 months compared to at 2 months (*Figure 7A–B*). Further, the number of close proximity potential contact sites with IBA1-labeled microglia increased with age, while proximity between R cells and OPCs remained unchanged (*Figure 7A–B*).

To further confirm the findings of the randomized microniche analyses, we took a more targeted approach, measuring the volumetric distribution of niche cells radiating out from Nestin-labeled radial processes and KI67-labeled proliferating cells (*Figure 7C*). Consistent with R cell contact analyses, we found reduced volumes of blood vessels in the immediate vicinity (<15 μm distance) of radial processes with densities being unchanged at further distances in 6-month-old animals. Vascular densities were comparable between 2- and 12-month-old animals across all distances from R cells (*Figure 7D*). In contrast, the relative volume of microglia around R cell processes was increased across all radii in 6- and 12-month-old mice (*Figure 7C*). For the environment of proliferating cells, we found that vascular density was markedly reduced in aged animals, while microglia were unchanged across all distances (*Figure 7C*), suggesting that IBA1-expressing microglia is important to maintain the niche but does not directly affect cell proliferation of neurogenic cells within the DG.

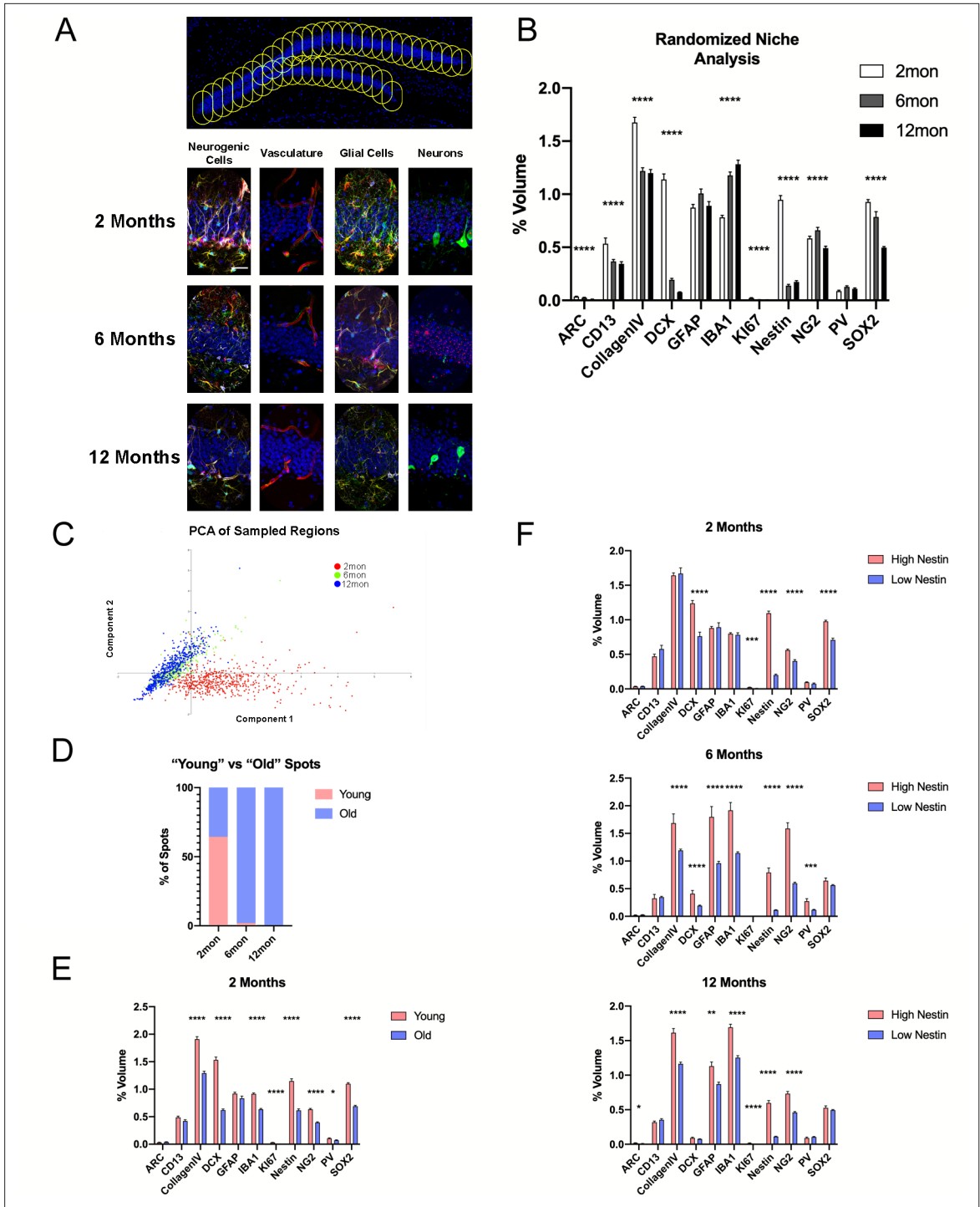

**Figure 6.** Microniche analysis reveals features of the neurogenic niche with advancing age. (**A**) A representative overview of the arrangement of randomized sampling spots, with close-up examples of the cellular contents within a single sampled spot for each age group. (**B**) Quantification of volumes of cell types within the random sampled spots normalized to the sampled area. For details of statistics please refer to *Supplementary file 1*. (**C**) Principle component analysis for dimensional reduction and clustering of the random sampled spots from 2-, 6-, and 12-month groups based on cellular content. (**D**) Percentage of spots identified as either 'young' or 'old' separated through k-means analysis. (**E**) Comparison of cell volumes in spots identified as young and old in the 2-month-old group. (**F**) Comparison of cell volumes in spots identified as high and low R cell populated as neurogenic classified of age in the 2- (upper graph), 6- (middle graph), and 12-month group (lower graph). Scale bars represent 25 μm (**A**, lower panels) and 100 μm (upper panel). *p<0.05, **p<0.01, ***p<0.001, ****p<0.0001.

The online version of this article includes the following figure supplement(s) for figure 6:

**Figure supplement 1.** Quantification of proliferating and non-proliferating cellular microenvironments in the DG.

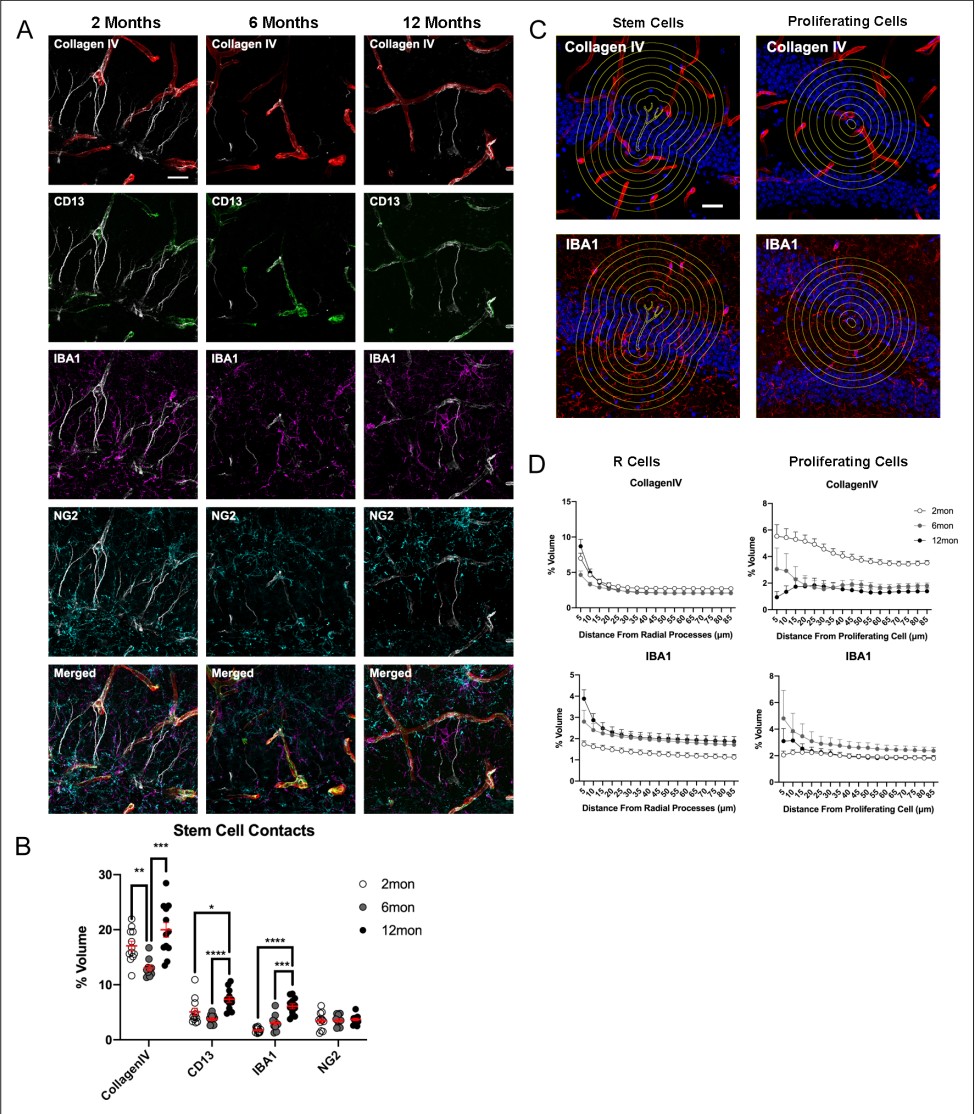

**Figure 7.** Age-dependent dynamics of the niche surrounding adult stem cells. (**A**) Representative images showing the interactions of R cells with vascular and glial cells in the surrounding niche in the DG in 2-, 6-, and 12-month-old mice. (**B**) Quantification of colocalization of blood vessels, pericytes, microglia, and OPCs with CollagenIV+ represented a percentage of total vascular volume. For details of statistics please refer to *Supplementary file 1*. (**C**) Representative images of areas measured for volumes of vasculature and microglia at radiating distances around Nestin+ R cells and Ki67+ proliferating cells. (**D**) Quantification of volumes of vasculature and microglia at radiating distances from R cells and Ki67+ proliferating cells. For details of statistics please refer to *Supplementary file 1*. Scale bars represent 25 µm. Scale bar represents 100 µm. *p<0.05, **p<0.01, ***p<0.001, ****p<0.0001. DG, dentate gyrus; OPC, oligodendrocyte precursor cell.

## Discussion

We here used 4i on tissue sections with the aim to characterize age-related changes in the number and distribution of a variety of cell types within individual tissue sections of the DG. We show that alterations within the neurogenic niche of the DG occur relatively early in life and identify changes in niche cell composition of the BBB and microenvironments surrounding adult NSCs as putative regulators of neurogenic permissiveness throughout life.

Reduced levels of neurogenesis in the mammalian DG have been associated with age-related cognitive decline and neurodegenerative diseases, such as Alzheimer's disease (*Kuhn et al., 1996*; *Drapeau et al., 2003*; *Ben Abdallah et al., 2010*; *Knoth et al., 2010*; *Boldrini et al., 2018*; *Moreno-Jiménez et al., 2019*; *Tobin et al., 2019*; *Denoth-Lippuner and Jessberger, 2021*). Thus, previous

work aimed to identify age-dependent changes within neurogenic NSCs and their surrounding niches. A number of cell-intrinsic and extrinsic factors have been identified that appear to mediate the age-related decline in NSC activity (*Verbitsky et al., 2004*; *Hafezi-Moghadam et al., 2007*; *Leeman et al., 2018*; *Dulken et al., 2019*; *Martín-Suárez et al., 2019*; *Morrow et al., 2020*; *Vonk et al., 2020*; *Bedrosian et al., 2021*). Further, structural but also functional benefits have been associated with experimentally enhanced neurogenesis in mice of advanced age (*Villeda et al., 2011*; *Katsimpardi et al., 2014*; *McAvoy et al., 2016*; *Fan et al., 2017*; *Ozek et al., 2018*; *Berdugo-Vega et al., 2020*; *Horowitz et al., 2020*). A critical role for vasculature within the dense vessel network of the DG has been described (*Palmer et al., 2000*; *van Praag et al., 2005*; *Shen et al., 2019*). However, it remained difficult to obtain a comprehensive view of age-related changes within the DG niche due to the difficulty to combine decisive cellular phenotyping of neurogenic cells with a number of different cell type markers within the same tissue section, a requirement to obtain a more complete picture of what changes indeed occur within or in the direct vicinity of NSCs in the mammalian DG. Using 4i, we here show that prominent changes in vascular microarchitecture and contact sites occur early on in life. Interestingly, we found that stem cells showed enhanced proximity with pericytes with advancing age. Further, our data indicate that both vascular density but also relative distribution of IBA1-labeled microglia and NG2-labeled oligodendroglia are associated with stem cell maintenance in the aged DG. If those cellular alterations are truly responsible for age-related changes of neurogenesis remain to be tested. Furthermore, additional alterations of niche cells may have been missed as the set of markers used to identify non-neurogenic cells in the DG do not indicate cell states or subtypes. Future experiments will aim to selectively manipulate newly identified aspects of niche interactions and composition that were identified here, probing for their functional significance. Together with other omics-based approaches that aim to identify age-dependent changes in the aging brain and within NSCs and their progeny (*Shavlakadze et al., 2019*; *Ximerakis et al., 2019*; *Ibrayeva et al., 2020*), the 4i-based data set we provide here will contribute to future work with the aim to enhance neurogenesis in aged brains. 4i-based data will contribute to such efforts by not only allowing for high spatial resolution of obtained expression data within complex tissues but also allowing for the characterization of interactions in the 3D space.

We here used 18 different antibodies to label identical tissue sections. However, that number may be easily scaled up when required. We show that 4i is applicable to a variety of mouse and human tissues, indicating its potential use in basically all tissues and organs. Thus, 4i will be useful to analyze a plethora of tissues and may turn out to be especially useful when only limited tissues/cells are available. For instance, it is a current fundamental limitation in clonal lineage tracing experiments that daughter cells cannot be definitely phenotyped. Rather, current limitations only allow for exclusion-based phenotyping, at least when starter/mother cells can generate a diverse set of daughter cell fates (e.g., a cell expresses protein X and Y but not Z; because only X and Y can be combined for analysis within a single stain, the negativity of Z cannot be shown and phenotypic classification needs to rely on exclusion or non-standardized morphological criteria). Thus, to date, all clonal lineages that generate >2 cell types can only be partially phenotyped and diverse cell types cannot be conclusively classified within the exact same clone (*Bonaguidi et al., 2011*; *Pilz et al., 2018*). 4i on tissue sections will overcome these technical limitations in an easy and readily applicable way, allowing for the use of extensively characterized and validated sets of antibodies with indirect immunofluorescence.

We here demonstrate the applicability of iterative immunostaining in tissue sections using 4i, provide a comprehensive data set of the aging DG with a large set of combined markers, and identify vasculature- and glia-associated changes within the neurogenic niche that may be responsible to mediate the age-related decline of neurogenesis in the mammalian hippocampus.

## Materials and methods

### Key resources table

| Reagent type (species) or resource | Designation | Source or reference | Identifiers | Additional information |
|---|---|---|---|---|
| Biological samples (*Mus musculus*) | C57BL/6JRj | Janvier Labs | | https://www.janvier-labs.com/en/fiche_produit/c57bl-6jrj_mouse/ |

*Continued on next page*

| Reagent type (species) or resource | Designation | Source or reference | Identifiers | Additional information |
|---|---|---|---|---|
| Antibody | ARC (Guinea pig Polyclonal) | Synaptic Systems | RRID:AB_2619853 | Dilution: (1:500) |
| Antibody | BLBP (Rabbit Polyclonal) | Abcam | RRID:AB_880078 | Dilution: (1:200) |
| Antibody | CD13 (Goat Polyclonal) | Novus | RRID:AB_2227288 | Dilution: (1:500) |
| Antibody | Cleaved Caspase 3 (Rabbit Polyclonal) | Cell Signaling Technology | RRID:AB_2341188 | Dilution: (1:150) |
| Antibody | CollagenIV (Rabbit Polyclonal) | Bio-Rad | RRID:AB_2082660 | Dilution: (1:750) |
| Antibody | CTIP2 (Rat Monoclonal) | Abcam | RRID:AB_2064130 | Dilution: (1:250) |
| Antibody | Doublecortin (Goat Polyclonal) | Santa Cruz | RRID:AB_2088491 | Dilution: (1:300) |
| Antibody | GFAP (Chicken Polyclonal) | Novus | RRID:AB_1556315 | Dilution: (1:750) |
| Antibody | Histone H3 Phospho S10 (Mouse Monoclonal) | Abcam | RRID:AB_443110 | Dilution: (1:250) |
| Antibody | HOPX (Mouse Monoclonal) | Santa Cruz | RRID:AB_2687966 | Dilution: (1:500) |
| Antibody | IBA1 (Rabbit Polyclonal) | WAKO | RRID:AB_839504 | Dilution: (1:500) |
| Antibody | ID4 (Rabbit Monoclonal) | Biocheck | RRID:AB_2814978 | Dilution: (1:250) |
| Antibody | KI67 (Rat Monoclonal) | Bioscience | RRID:AB_10854564 | Dilution: (1:250) |
| Antibody | LaminB1 (Rabbit Polyclonal) | Abcam | RRID:AB_10107828 | Dilution: (1:500) |
| Antibody | MT3 (Rabbit Monoclonal) | Abcam | RRID:AB_2297959 | Dilution: (1:300) |
| Antibody | Nestin (Mouse Monoclonal) | BD | RRID:AB_396354 | Dilution: (1:250) |
| Antibody | NeuroD1 (Goat Polyclonal) | Santa Cruz | RRID:AB_630922 | Dilution: (1:250) |
| Antibody | NG2 (Rabbit Polyclonal) | Millipore | RRID:AB_11213678 | Dilution: (1:500) |
| Antibody | OLIG2 (Rabbit Polyclonal) | Millipore | RRID:AB_570666 | Dilution: (1:500) |
| Antibody | Parvalbumin (Mouse Monoclonal) | Sigma-Aldrich | RRID:AB_477329 | Dilution: (1:250) |
| Antibody | S100β (Rabbit Monoclonal) | Abcam | RRID:AB_882426 | Dilution: (1:500) |
| Antibody | SOX2 (Rat Monoclonal) | Invitrogen | RRID:AB_11219471 | Dilution: (1:250) |
| Antibody | Anti-Chicken IgY (IgG) 488 (Donkey Polyclonal) | Jackson Laboratory | RRID:AB_2340375 | Dilution: (1:250) |
| Antibody | Anti-Chicken IgY (IgG) 647 (Donkey Polyclonal) | Jackson Laboratory | RRID:AB_2340379 | Dilution: (1:250) |

| Reagent type (species) or resource | Designation | Source or reference | Identifiers | Additional information |
|---|---|---|---|---|
| Antibody | Anti-Goat IgG 488 (Donkey Polyclonal) | Jackson Laboratory | RRID:AB_2336933 | Dilution: (1:250) |
| Antibody | Anti-Goat IgG 647 (Donkey Polyclonal) | Jackson Laboratory | RRID:AB_2340437 | Dilution: (1:250) |
| Antibody | Anti-Goat IgG Cy3 (Donkey Polyclonal) | Jackson Laboratory | RRID:AB_2307351 | Dilution: (1:250) |
| Antibody | Anti-Guinea pig IgG488 (Donkey Polyclonal) | Jackson Laboratory | RRID:AB_2340472 | Dilution: (1:250) |
| Antibody | Anti-Guinea pig IgG647 (Donkey Polyclonal) | Jackson Laboratory | RRID:AB_2340476 | Dilution: (1:250) |
| Antibody | Anti-Mouse IgG 488 (Donkey Polyclonal) | Jackson Laboratory | RRID:AB_2341099 | Dilution: (1:250) |
| Antibody | Anti-Mouse IgG 647 (Donkey Polyclonal) | Jackson Laboratory | RRID:AB_2340863 | Dilution: (1:250) |
| Antibody | Anti-Mouse IgG Cy3 (Donkey Polyclonal) | Jackson Laboratory | RRID:AB_2315777 | Dilution: (1:250) |
| Antibody | Anti-Rabbit IgG 488 (Donkey Polyclonal) | Jackson Laboratory | RRID:AB_2313584 | Dilution: (1:250) |
| Antibody | Anti-Rabbit IgG 647 (Donkey Polyclonal) | Jackson Laboratory | RRID:AB_2492288 | Dilution: (1:250) |
| Antibody | Anti-Rabbit IgG Cy3 (Donkey Polyclonal) | Jackson Laboratory | RRID:AB_2307443 | Dilution: (1:250) |
| Antibody | Anti-Rat IgG 488 (Donkey Polyclonal) | Jackson Laboratory | RRID:AB_2340684 | Dilution: (1:250) |
| Antibody | Anti-Rat IgG 647 (Donkey Polyclonal) | Jackson Laboratory | RRID:AB_2340694 | Dilution: (1:250) |
| Antibody | Anti-Rat IgG Cy3 (Donkey Polyclonal) | Jackson Laboratory | RRID:AB_2340667 | Dilution: (1:250) |
| Chemical compound, drug | Donkey serum | Millipore | Cat#: 530-100 ML | |
| Chemical compound, drug | Ethylene glycol | Sigma-Adrich | Cat#: 324558 | |
| Chemical compound, drug | Glycerol | Sigma-Aldrich | Cat#: G5516 | |
| Chemical compound, drug | Glycine | Biosolve | Cat#: 071323 | |
| Chemical compound, drug | Guanidinium chloride | Sigma-Aldrich | Cat#: G4505 | |
| Chemical compound, drug | Hydrochloric acid standard 33% solution | Sigma-Aldrich | Cat#: 71826 | |

| Reagent type (species) or resource | Designation | Source or reference | Identifiers | Additional information |
|---|---|---|---|---|
| Chemical compound, drug | N-Acetyl-Cysteine | Sigma-Aldrich | Cat#: A9165 | |
| Chemical compound, drug | NaCl | Sigma-Aldrich | Cat#: S9625 | |
| Chemical compound, drug | O.C.T compound TissueTek | Sakura | Cat#: 4583 OCT 25608-930 | |
| Chemical compound, drug | Para formaldehyde | Sigma-Aldrich | Cat#: 441244 | |
| Chemical compound, drug | Poly-D-Lysine | Sigma-Aldrich | Cat#: P6407 | |
| Chemical compound, drug | Potassium chloride | Sigma-Aldrich | Cat#: P9333 | |
| Chemical compound, drug | Potassium phosphate monobasic | Sigma-Aldrich | Cat#: P8709 | |
| Chemical compound, drug | Sodium hydroxide | Sigma-Aldrich | Cat#: S5881 | |
| Chemical compound, drug | Sodium phosphate dibasic dehydrate | Sigma-Aldrich | Cat#: 30435 | |
| Chemical compound, drug | Sodium phosphate monobasic monohydrate | Sigma-Aldrich | Cat#: S9638 | |
| Chemical compound, drug | Sucrose | Sigma-Aldrich | Cat#: 84100 | |
| Chemical compound, drug | TCEP-HCl | Sigma-Aldrich | Cat#: C4706 | |
| Chemical compound, drug | Triton X-100 | Sigma-Aldrich | Cat#: 93443 | |
| Chemical compound, drug | Urea | Sigma-Aldrich | Cat#: U1250 | |
| Software, algorithm | GraphPad Prism | GraphPad | RRID:SCR_002798 | http://www.graphpad.com/ |
| Software, algorithm | Fiji/ImageJ | Fiji | RRID:SCR_002285 | http://fiji.sc |
| Software, algorithm | ZEN Blue | Carl Zeiss AG | RRID:SCR_013672 | http://www.zeiss.com/microscopy/en_us/products/microscope-Software, Algorithm/zen.html#introduction |
| Software, algorithm | Past4.03 | Oyvind Hammer | RRID:SCR_019129 | http://folk.uio.no/ohammer/past/ |
| Other | DAPI | Sigma-Aldrich | Cat#: D9542 | Dilution: (1:1000) |

## Mice

Animal experiments were approved by the Cantonal Commission for Animal Experimentation of the Canton of Zurich, Switzerland in accordance with national and cantonal regulations (license ZH037/2017; ZH126/2020). In developing the 4i protocol for tissue sections, brain tissue was collected from male and female C57Bl/6 mice ranging between 2 and 4 months of age. To attain embryonic day (E) 14.5 embryonic samples, daily plug checks were used to estimate the date of conception. Embryos were collected 14 days following the observation of a vaginal plug. For the aging study, C57BL/ 6J males at 2 (n=3), 6 (n=3), and 12 months (n=3) of age were used. All mice were kept in a 12 hr light/ dark cycle with food and water provided ad libitum.

## Tissue preparation

For adult tissues, animals were transcardially perfused using a peristaltic pump with cold 0.9% saline solution followed by 4% paraformaldehyde (PFA) in phosphate buffer. Brains were collected and post-fixed in 4% PFA at 4°C for 8 hr. The brains were then cryo-protected in 30% sucrose for 2 days prior to sectioning. Adult mouse brain tissue was sectioned into 40 µm coronal sections on a cryotome (Leica SM2010R). For E14.5 samples, embryonic heads from C57Bl/6 mice were washed in phosphate-buffered saline (PBS) and fixed with 4% PFA overnight. Cryoprotection occurred in two steps, first for 2 days in 15% sucrose followed by 2 days in 30% sucrose, before being flash-frozen and embedded in OCT Compound (Tissue-Tek) using liquid nitrogen. Sectioning was done coronally at 40 µm using a cryostat. Sections were collected and washed with PBS to remove the remaining OCT before being transferred to cryoprotectant solution (CPS, 25% Ethyleneglycol, 25% Glycerin, and 0.1 M phosphate buffer). Forebrain-specific organoids derived from hESCs (approved by the Kantonale Ethikkomission Zurich and reported to the federal Bundesamt für Gesundheit of Switzerland) were generated as described before (*Bowers et al., 2020*), and fixed at day 40 in 4% PFA for 15 min, embedded in OCT, and flash-frozen using liquid nitrogen. Organoids were sectioned at 40 µm using a cryostat, sections were collected in PBS. Serial washes with PBS were performed to remove the remaining OCT, before being transferred to CPS.

## Mounting

Prior to mounting, glass-bottomed 24-well plates (Cellvis P24-1.5H-N) were coated with 1 mg/ml PDL (Sigma-Aldrich P6407). 75 µl of the PDL solution was transferred into the center of each well and brushed to the edges with a fine paintbrush to ensure total coverage. The plates were rocked on a shaker for 5 min, after which the PDL was collected for reuse. Wells were rinsed three times with deionized water and let dry for at least 2 hr. Tissue sections were washed three times in PBS, loaded into wells containing 500 µl 1× PBS, which was subsequently aspirated allowing the tissue to lie flat and dry for approximately 20 min until there was no visible liquid remaining around the edges of the sections. The tissue was then rinsed for 30 s with 4% PFA and washed in PBS three times for 5 min each. Organoids sections were gently aspirated using a P1000 pipet with a trimmed tip and transferred into a well of a glass bottomed well plate. The sections were allowed to settle onto the glass before carefully removing the excess liquid and allowing to dry. Mounted organoid sections were additionally fixed with 4% PFA for 15 min followed by three washes with PBS. Prior to staining, the organoid sections were washed with the elution buffer three times for 5 min as a form of mild antigen retrieval.

## Staining

Once mounted, sections were blocked for 1 hr at room temperature in a blocking solution containing PBS with 3% donkey serum, 0.5% Triton-X, and 0.025% PFA (PBS++). Sections were incubated in the primary antibodies (*Supplementary file 1*), diluted in the blocking solution, for between two to four nights, depending on the efficacy of the antibodies, shaking at 4°C. Following the primary antibody incubations, the tissue was washed two times with PBS for 5 min and then blocked with PBS++ for 1 hr at room temperature. Sections were incubated with secondary antibodies, diluted at a ratio of 1:250 along with DAPI at 1:1000 in PBS++, for 3 hr shaking at room temperature, after which the tissue was washed three times with PBS for 5 min. Once the secondary antibody solutions were added all liquid handling was conducted in low light until the antibodies were eluted to reduce the chance

of fluorophore crosslinking. The imaging buffer was added at least 5 min prior to imaging to ensure the penetrance of the tissue.

## Imaging

Before imaging, fresh imaging buffer was prepared by dissolving 0.7 M N-Acetyl-Cysteine in 0.2 M phosphate buffer, using 10 M NaOH to adjust the pH to 7.40. Approximately 500 µl of the imaging buffer was added to each well. All 4i images were taken on a Zeiss LSM800 confocal laser-scanning microscope. Images from tile scans were exported using Carl Zeiss ZEN Blue software and stitched using Fiji (*Schindelin et al., 2012*). For the aging study, DG image stacks were taken with a 20× magnification objective with a 0.8 numerical aperture and composed of tile regions with 10 tiles (5×2), with a pixel resolution of 1024×1024 per tile for a scale of 3.2055 pixels/µm with 16-bit pixel depth. Image stacks consisted of 40 frames acquired with a z-step interval of 1 µm and were used for all quantifications. High-resolution close-ups were acquired with a 40× magnification objective with a 1.1 numerical aperture in a water immersion (Zeiss Immersol W 2010) at a pixel resolution of 2048×2048 with a z-step size of 0.5 µm for 80 frames. These 40× images were used as example images in figures. All acquisitions were done with bi-directional scanning with a pixel dwell time of 2.06 µs. Within each cycle, all samples were labeled with the same antibodies, and imaged with identical microscopy settings for laser power, gain, digital offset, pinhole diameter, and z-step.

## Elution

A stock solution containing 0.5 M L-Glycine, 3 M Urea, and 3 M Guanidine hydrochloride was prepared and kept at 4°C. Prior to elution, 0.07 M (0.02 g/ml) TCEP-HCl was added to a volume of the stock solution determined by multiplying the number of wells by the working volume of 150 µl by three washes. The pH of the buffer was then lowered to 2.5 using 5 M HCl to facilitated denaturation of the antibodies. The tissue was first rinsed with dH$_2$O, then 150 µl of dH$_2$O followed by 150 µl of the elution buffer was added to each well, shaken for 5 min, and repeated for a total of three washes. After elution, the tissue was washed three times for 5 min in PBS, after which the blocking step for the next round of IF could be started.

## Image alignment

To achieve accurate alignment between rounds of staining, images were registered using consistent DAPI intensity patterns. At a high digital zoom, the x and y coordinates, as well as the stack position of pixels in DAPI puncta that were recognizable across cycles and were measured. Images were transformed in the x and y planes to align to the image from the first round, with the canvas size being adjust so no image cropped and data was lost. For alignment in the z-plane, blank frames were added or frames lacking positive fluorescent signal were subtracted from the image stack accordingly using the 'Add Slice' and 'Delete Slice' functions in ImageJ.

## Antigenicity test

Two brain sections from 2-month-old mice were mounted and stained for all 18 antibodies used in the main study. Regions of the DG were imaged with a Zeiss LSM800 confocal laser-scanning microscope at 20× with the same parameters as in the main study. The sections were then subjected to six rounds of elution washes (3×5 min). The brain sections were incubated with the same secondary antibodies used in the first round and re-imaged to confirm successful antibody removal. A subsequent cycle of staining and imaging was performed with the same primary and secondary antibody combinations as in the first round (*Supplementary file 2*). Images across cycles were acquired using the same microscope settings (laser power, gain, digital offset, pinhole diameter, and z-step). Regions of interest (ROIs) were drawn around positive cells in images from the first cycle and fluorescent intensities were measured in the same cells from the two rounds of staining as well as following elution and were correlated. For KI67, phosphorylated histone-3, and parvalbumin, too few cells were present in the sections to correlate values.

## Image analysis

All quantifications were performed on non-projected 3D z-stack images with cells being sampled throughout the entire thickness of the image. All measures of density and intensity were performed

on 16-bit images, while volumetric analyses were performed using 8-bit binary representations of the original images.

Cell counting: R cells were defined by HOPX$^+$/SOX2$^+$ soma with a clear vertical process that was also GFAP$^+$ and S100β$^-$. ROIs were created outlining the cell soma using the polygon selection tool in Fiji, which were saved and later used to measure fluorescent intensities of stem cell markers HOPX, and MT3, the intensity of the nuclear envelope protein LaminB1, as well as checking for the presence of the cell cycle marker KI67. NR glia-like cells in cell cycle were identified by being KI67$^+$ and located within the lower granular cell layer or SGZ extending approximately 20 µm beyond the hilar edge of the GCL. Immature neurons were counted using DCX. Positive cells had a clear ring of non-nuclear staining that encircled a significant majority of the cell (approx. 75%). Glial cell types, astrocytes, microglia, and oligodendrocytes, were identified as being S100β$^+$, IBA1$^+$, or NG2+, respectively. All cells were counted throughout the entire images using the Cell Counter plugin in Fiji using localization identifiers for the ML, hilus/CA3, and the supra and infrapyramidal blades of the GCL. The areas of the hippocampal subregions were measured. Cell density was calculated by dividing the number of cells by the regional volume expressed as mm$^3$ (region area [mm$^2$] × tissue thickness [0.04 mm]).

Fluorescent intensity was measured in the z-position in which it was brightest for each cell. Measures were normalized to the background intensity, which was averaged from five ROIs per slice containing no positive cells within the GCL. Cells were normalized to the background of the representative slice in which the cells were measured to account for any gradient in fluorescent intensity caused by unequal antibody penetrance. The fluorescent intensity of LaminB1 was measured in R cells, immature neurons, and PV interneurons using the ROIs generated during cell counting. For glial cell types, 100 cells were selected at random throughout the imaged hippocampal structures and somatic ROIs were drawn. To ensure accurate measurement, if necessary, ROIs were adjusted to better fit the LaminB1 outline. Intensity measures were normalized to the background, which was averaged from five ROIs per slice throughout the tissue containing no positive signal. Like with the RGL markers, normalization was slice specific to account for a gradient in intensity across the z-plane. The ROIs made to measure LaminB1 in S100β$^+$ astrocytes were also used to measure the intensities of HOPX and MT3 in astrocytes.

Vasculature dimensions: All measures of vasculature dimensions were conducted using CollagenIV staining within the visible areas of the hippocampus (areas of the thalamus were excluded). To calculate vasculature density, volumes were generated based on Cavalieri's principle, summing the area of blood vessels in each slice of an image stack and dividing by the tissue volume. Vasculature areas were measured by applying a 3D Gaussian blur with pixel radii of 2, 2, and 1.5 in the x, y, and z directions, respectively, thresholding at a fluorescence intensity of 1500, followed by two binary erosions to generate an accurate binary representation of the CollagenIV staining.

Cellular interactions: Cellular interactions were estimated by measuring the area of colocalization of cell markers on the vasculature and RGL processes. RGL processes were manually isolated from vasculature in Nestin stains by outlining Nestin$^+$ RGL processes into an ROI and clearing the outside. Binary representations were created the same as was done for CollagenIV for CD13, GFAP, and the isolated Nestin processes with thresholding at lower intensity limits of 1800, 5250, and 2000, respectively. For NG2, and IBA1, a median filter with a radius of 2 pixels was applied and thresholds were applied at 5500 and 2000, respectively. ROIs were generated by creating selections around the binary CollagenIV and Nestin, which were then applied to the other stains, measuring the area of positive pixels within the ROIs. The measured areas were normalized to the total ROI areas to get percent of coverage. S100β was excluded from all contact and volumetric analyses as the level of background prevented the creation of accurate binary representations.

Randomized niche mapping: Binary representations of DCX, HOPX, SOX2, PV, ARC, and KI67 were generated in the same manner as CollagenIV with thresholds set at 2500, 2200, 1600, 4300, 2500, and 2000, respectively. In generating binaries with intensity thresholding, Arc and Ki67 images had small false positive puncta and required additional processing to remove these background elements. Particle analysis was run to select all for positive cells using a lower area limit of 25 µm for ARC, and 15 µm for Ki67. ROIs were saved and used to draw the cells onto a blank canvas. To generate randomized sample regions, ROIs were created around the GCL using maximum z-projections of DAPI stains, which were then overlaid and filled in white on a blank image. A second equivalent blank image was subdivided by drawing vertical white bars extending the height of the canvas with a width of 1 pixel,

spaced 160 pixels apart, equivalent to 50 μm in the microscope image scale. The filled representation of the GCL was multiplied with the image containing the bars resulting in an image with white bars spaced 160 pixels apart, fit to the shape and spanning the width of the blades of the GCL. ROIs were created for each bar individually by particle analysis. On a blank canvas, the bars were filled and an intensity gradient distance map was generated around it. The pixels in the bar had an intensity value of 0, with intensities increasing by one with each pixel radiating away from the origin until a value of 255. The binary representations of ARC, CD13, CollagenIV, DCX, GFAP, HOPX, IBA1, Ki67, Nestin, NG2, PV, SOX2, and S100β, which were divided by 255 so the new intensity value was 1. The binary distance maps for each bar were multiplied against the divided binary images created an image with 8-bit intensity profile representative of the distance from the point of origin. Analyzing the histogram of the resulting images provided the areas and spatial distribution of positive signal for each stain within the distance map. Areas of stains were summed within the distance maps to up to a 160 pixel (50 μm) radius, which provided adequate and contiguous sampling of the GCL, bordering areas of the hilus, and inner ML. The areas were also summed across the z-stack creating volumes. These were normalized by dividing by the total volume sampled by the distance map and reported as a percentage.

Targeted niche mapping: Distance maps were generated around the Nestin[+] R cell processes, and proliferating cells, and measured against the binary representations of cells the same as was done for the randomized probing. Nestin-labeled R cell processes were measured as a whole per section. Ki67[+] cells were isolated using particle analysis on the maximum intensity z-projection. ROIs were created around the cells and individually applied to the 3D binary to clear signal outside of the cell of interest. Stain volumes were binned at 5 μm intervals and divided by the total measured volume and reported as percent coverage.

## Statistical analyses

For all analyses, an alpha level was set at 0.05. To assess preservation of antigenicity Pearson correlations were performed on the normalized intensity values of isolated cells or ROIs. Unpaired t-tests were used to compare counted cell densities, as well as quiescent cell marker and LaminB1 intensities of DCX-labeled immature neurons and PV interneurons between the 2- and 12-month groups. The standard deviations between age groups were significantly different for the intensities of LaminB1 in R cells, IBA1-labeled microglia, and S100β[+] astrocytes so they were compared using Welch's t-tests. One-way ANOVAs were performed to assess differences between ages in vascular densities, blood vessel coverage, randomized microniche volumes, and stem cell contact measures. These analyses were followed by post hoc Tukey's multiple comparisons tests. When data failed the Brown-Forsythe test for equal variances, Welch's ANOVA tests were run with Dunnett's T3 multiple comparisons. For the microniche volumetric measures, Games-Howell's multiple comparisons were used to account for the large group sizes. In comparing spots identified as young and old, Nestin high and low, and KI67 positive and negative, multiple unpaired t-tests were performed with Holms-Sidak correction for multiple comparisons. Microenvironments surrounding radial processes and proliferating cells were analyzed using mixed-effects model analysis and Tukey's multiple comparisons tests.

## Acknowledgements

The authors thank P Eugster for analytical help, G Pilz for conceptual input and advice, and A Denoth-Lippuner and DC Lie for comments on the manuscript.

## Additional information

### Competing interests
Gabriele Gut, Lucas Pelkmans: Inventor of a patent related to 4i technology (WO2019207004A1). The other authors declare that no competing interests exist.

## Funding

| Funder | Grant reference number | Author |
|---|---|---|
| European Research Council | STEMBAR | Sebastian Jessberger |
| Schweizerischer Nationalfonds zur Förderung der Wissenschaftlichen Forschung | BSCGI0_157859 | Sebastian Jessberger |
| Schweizerischer Nationalfonds zur Förderung der Wissenschaftlichen Forschung | 310030_196869 | Sebastian Jessberger |
| European Research Council | ERC-2019-AdG-885579 | Lucas Pelkmans |
| Schweizerischer Nationalfonds zur Förderung der Wissenschaftlichen Forschung | 310030_192622 | Lucas Pelkmans |

The funders had no role in study design, data collection and interpretation, or the decision to submit the work for publication.

## Author contributions

John Darby Cole, Data curation, Formal analysis, Investigation, Methodology, Visualization, Writing – original draft; Jacobo Sarabia del Castillo, Gabriele Gut, Daniel Gonzalez-Bohorquez, Methodology; Lucas Pelkmans, Conceptualization, Funding acquisition, Supervision; Sebastian Jessberger, Conceptualization, Funding acquisition, Project administration, Resources, Writing – original draft

## Author ORCIDs

John Darby Cole ⬚ http://orcid.org/0000-0002-3615-8702
Gabriele Gut ⬚ http://orcid.org/0000-0001-8991-0040
Lucas Pelkmans ⬚ http://orcid.org/0000-0002-6754-9730
Sebastian Jessberger ⬚ http://orcid.org/0000-0002-0056-8275

## Ethics

Animal experiments were approved by the Cantonal Commission for Animal Experimentation of the Canton of Zurich, Switzerland in accordance with national and cantonal regulations (license ZH037/2017; ZH126/2020).

## Decision letter and Author response

Decision letter https://doi.org/10.7554/eLife.68000.sa1
Author response https://doi.org/10.7554/eLife.68000.sa2

# Additional files

## Supplementary files

• Supplementary file 1. Values and statistics for graphs shown in main *Figures 1 and 3–7* and *Figure 6—figure supplement 1* of *Figure 6*.
• Supplementary file 2. Primary and secondary antibody combinations used for antigenicity testing.
• Transparent reporting form

## Data availability

Data was deposited to the Image Data Resource under accession number idr0131.

The following dataset was generated:

| Author(s) | Year | Dataset title | Dataset URL | Database and Identifier |
|---|---|---|---|---|
| Cole JD, Jessberger S | 2022 | Data from:Characterization of the neurogenic niche in the aging dentate gyrus using iterative immunofluorescence imaging | https://idr. openmicroscopy. org/search/?query= Name:131 | Image Data Resource, idr0131 |

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
