## [Decision Letter]

**Acceptance summary:**

This paper describing the development of a new anatomical method in the context of adult hippocampal neurogenesis in the aging brain will have a high impact for the scientific community. It will allow to deepen our understanding of cellular complexity in the normal and disease brain.

**Decision letter after peer review:**

Thank you for submitting your article "Characterization of the neurogenic niche in the aging dentate gyrus using iterative immunofluorescence imaging" for consideration by *eLife*. Your article has been reviewed by 3 peer reviewers, and the evaluation has been overseen by a Reviewing Editor and Marianne Bronner as the Senior Editor. The following individual involved in review of your submission has agreed to reveal their identity: Tim James Viney (Reviewer #2).

Essential revisions:

The reviewers are very enthusiastic about your work and the impact of this 4i method for the scientific community. However, more information is needed to improve the manuscript. In particular, we recommend to

i) Strengthen the reporting of the methods,

ii) Add some control for the consistency in antibody staining , the sensitivity of the elution process, individual section variability etc.

iii) To provide high resolution single-channel images.

Concerning the main criticism of Reviewer 1, you may as requested add a group of old mice (>18 months) or alternatively remove all references to aging in the title, abstract and discussion.

In addition, it would be very beneficial for the community to add in the discussion (or methods/results) comments about the boundaries conditions of the 4i method (pitfall, what not to do…).

*Reviewer #1 (Recommendations for the authors):*

1) This study does not addresses ageing but the transition of the hippocampal niche to an adult or mature stage, the text including the title should be modified to reflect this.

2) There is no evidence that staining for the different antibodies used works as well in the 2-month and 12-month-old sections (except for the similar expression levels of LaminB1 in some cell types), expression of a protein that is not expected to change between the two stages should be analysed alongside the proteins analysed, as a control of consistency in antibody staining.

3) In contrast with the thorough characterisation of neural stem cells, other cell types in the niche are identified with only one marker. This might reduce the ability of the authors to draw conclusions regarding age-related changes in the niche, particularly for cell types such as astrocytes or microglia that can exist in different subtypes or different activation states. This could be discussed.

4) The comparison of the intensity of antibody labelling before and after 6 rounds of elution (Figure 1E) is an important validation of the method. However it should be conducted systematically for each antibody used in 4i as different epitopes recognised by different antibodies may differ in their sensitivity to the elution process.

5) In the same figure, have the authors verified that not only the secondary antibody but also the primary antibody is eluted in each cycle (by re-incubating the eluted sections with the secondary antibody only)?

6) I find it somewhat counterintuitive that microniches from 2-month-old animals segregate into two groups of young-like and old-like spots rather than forming a continuum between these two states (Figure 6C). Please discuss the implications.

7) Proximity of radial processes with CollagenIV and CD13 diminishes between 2- and 6-months and increases between 6- and 12- months. There is therefore no clear trend in the changes that occur across the lifespan. This cast some doubt on the significance of the analysis and the results obtained. Please comment.

8) In Figure 7A-B, the authors conduct an analysis of contact sites of radial stem cells with other cell types in the niche. However the analysis does not seem to have enough spatial resolution to conclude that NSCs contact other niche cells rather than being in proximity but without contact. The conclusion from this analysis should be toned down.

*Reviewer #2 (Recommendations for the authors):*

I recommend strengthening the reporting of the methods, including examples of the specific analysis, and providing high resolution single-channel images demonstrating colocalisation and associations. This would significantly improve the ability to interpret the results and assess the conclusions.

---

## [Author Response]

Essential revisions:The reviewers are very enthusiastic about your work and the impact of this 4i method for the scientific community. However, more information is needed to improve the manuscript. In particular, we recommend toi) Strengthen the reporting of the methods,ii) Add some control for the consistency in antibody staining , the sensitivity of the elution process, individual section variability etc…iii) To provide high resolution single-channel images.Concerning the main criticism of Reviewer 1, you may as requested add a group of old mice (>18 months) or alternatively remove all references to aging in the title, abstract and discussion.In addition, it would be very beneficial for the community to add in the discussion (or methods/results) comments about the boundaries conditions of the 4i method (pitfall, what not to do…).

We thank the reviewing editor and the reviewers for their supportive and constructive comments on our manuscript. We are confident that our new experiments and analyses fully address the concerns previously raised by the reviewers. We feel that newly added data and analyses substantially strengthened our manuscript.

Reviewer #1 (Recommendations for the authors):1) This study does not addresses ageing but the transition of the hippocampal niche to an adult or mature stage, the text including the title should be modified to reflect this.

We understand the reviewer’s concern and have carefully edited our revised manuscript. We now carefully avoid the term “aged” when describing results for 12-month-old mice. We also more explicitly introduced what we refer to by the term “advancing age”. We had not meant to portray our findings in the context of “aged” mice (where we fully agree that older mice, e.g., 18-24 month-old, should have been used) but rather in the context of “advancing age”, which we define as functional alteration based on chronological age of an individual/tissue. Maturation until mid-life cannot be ruled out but given that in mice and humans function of cells seems to decline rather early on in life (certainly around the first third of the maximum life span if not much earlier), we do not believe that the changes in neurogenesis (dramatic decline of newborn neurons compared to sexually mature, young adult, 2-month-old mice) seen in 12-month-old mice purely reflect “maturation” of that brain area. We understand that the use of “aged” is not just semantics. However, we do feel that the use of the term “advanced age” is certainly justified and together with the introductory remarks now included will not mislead the reader.

2) There is no evidence that staining for the different antibodies used works as well in the 2-month and 12-month-old sections (except for the similar expression levels of LaminB1 in some cell types), expression of a protein that is not expected to change between the two stages should be analysed alongside the proteins analysed, as a control of consistency in antibody staining.

We agree with the reviewer and have now included novel data showing that

HOPX and MT3 – showing altered expression levels in neurogenic progenitors – do not change expression levels in classical astrocytes with age (please refer to revised Figure 4C). Furthermore, we do show that changes of LB1 with age are cell type specific (please refer to revised Figure 4E).

3) In contrast with the thorough characterisation of neural stem cells, other cell types in the niche are identified with only one marker. This might reduce the ability of the authors to draw conclusions regarding age-related changes in the niche, particularly for cell types such as astrocytes or microglia that can exist in different subtypes or different activation states. This could be discussed.

We understand the reviewer’s concern and have now extended the discussion accordingly. Please refer to page 15 of the revised manuscript.

4) The comparison of the intensity of antibody labelling before and after 6 rounds of elution (Figure 1E) is an important validation of the method. However it should be conducted systematically for each antibody used in 4i as different epitopes recognised by different antibodies may differ in their sensitivity to the elution process.

This is an excellent suggestion and we have now performed the suggested experiments. The new data are included in the revised extended data Figure 1. Furthermore, we have now added a more detailed description how “new” antibodies should be tested. Please refer to page 5 of the revised manuscript.

“To assess potential effects of cyclic staining and repeated elutions on sample antigenicity, adult brain sections were stained with the 18 antibodies used in the present study, imaged, and subsequently subjected to six rounds of elution prior to restaining. Measured fluorescence intensities between the stainings before and after rounds of elution were strongly correlated (Figure 1E and Figure 1 Supplemental), indicating that antigenicity is preserved across repeated cycles of iterative immunostainings for most of the antibodies that we used in tissue sections, similar to the high reproducibility that 4i achieves in cultured cells (Gut et al., 2018).”

5) In the same figure, have the authors verified that not only the secondary antibody but also the primary antibody is eluted in each cycle (by re-incubating the eluted sections with the secondary antibody only)?

We have now performed the suggested experiments, again corroborating the results that were obtained using cultured cells (Gut et al., 2018 Science). The new data are included in the revised extended data Figure 1.

6) I find it somewhat counterintuitive that microniches from 2-month-old animals segregate into two groups of young-like and old-like spots rather than forming a continuum between these two states (Figure 6C). Please discuss the implications.

Indeed, the microniches in 2-month-old mice do represent a continuum in a PCA (indicated by the stretch on component 1). However, there are already at the 2 months time point microniches that do resemble microniches in the DG of mice with more advanced age. We have discussed this finding now more explicitly. Please refer to page 10 and 11 of the revised manuscript. The split between young and old was manufactured by K-means clustering expecting two groups.

7) Proximity of radial processes with CollagenIV and CD13 diminishes between 2- and 6-months and increases between 6- and 12- months. There is therefore no clear trend in the changes that occur across the lifespan. This cast some doubt on the significance of the analysis and the results obtained. Please comment.

The reviewer is correct that stem cell contacts with CollagenIV are reduced at 6 months compared to 2-month-old mice and then increased between 6- and 12-month-old mice (CD13 is not different between 2- and 6-month-old mice). These are the data we observed that may indicate that the situation is distinct between 6- and 12-month-old mice. We do not understand how that result may “cast some doubt on the significance of the analysis”: it would be rather unlikely if the changes we observed would all follow a linear trajectory. However, we have extended the presentation of that finding and also followed the reviewer’s advice to tone down conclusions regarding contact sites (please refer to changes on page 11 and 12).

“To analyze possible interactions between R cells and other cell types in the niche, the area of cell marker co-localization (i.e., proximity) was measured. In general, a greater percentage of radial processes in 12-month-old animals were in close proximity with niche cell markers for vasculature, pericytes, and microglia, than in younger mice (Figure 7A-B). R cell processes were preferentially close to the vasculature in areas with low pericyte coverage (negative for CD13) (Licht et al., 2020), but co-localization of radial processes with CD13-labeled pericytes was increased at 12 months compared to at 2 months (Figure 7A-B). Further, the number of close proximity potential contact sites with IBA1-labeled microglia increased with age, while proximity between R cells and OPCs remained unchanged (Figure 7A-B).”

8) In Figure 7A-B, the authors conduct an analysis of contact sites of radial stem cells with other cell types in the niche. However the analysis does not seem to have enough spatial resolution to conclude that NSCs contact other niche cells rather than being in proximity but without contact. The conclusion from this analysis should be toned down.

We agree with the reviewer and have modified the description of our results accordingly. Please refer to page 11 and 12 of the revised manuscript (and to point 7).

Reviewer #2 (Recommendations for the authors):I recommend strengthening the reporting of the methods, including examples of the specific analysis, and providing high resolution single-channel images demonstrating colocalisation and associations. This would significantly improve the ability to interpret the results and assess the conclusions.

We understand the reviewer’s concerns and have carefully edited our manuscript to provide more methodological details. Further, we have added the single channels for the R Cell quantification (Supplemental Figure 3).